# The Complete Mitochondrial Genome of the Hermit Crab *Diogenes edwardsii* (Anomura: Diogenidae) and Phylogenetic Relationships within Infraorder Anomura

**DOI:** 10.3390/genes14020470

**Published:** 2023-02-12

**Authors:** Xiaoke Pang, Wenjing Fu, Jianfeng Feng, Biao Guo, Xiaolong Lin, Xueqiang Lu

**Affiliations:** 1Tianjin Key Laboratory of Environmental Technology for Complex Trans-Media Pollution and Tianjin International Joint Research Center for Environmental Biogeochemical Technology, College of Environmental Science and Engineering, Nankai University, Tianjin 300350, China; 2Department of Fishery Resources, Tianjin Fisheries Research Institute, Tianjin 300457, China; 3Shanghai Universities Key Laboratory of Marine Animal Taxonomy and Evolution, Shanghai Ocean University, Shanghai 201306, China; 4Engineering Research Center of Environmental DNA and Ecological Water Health Assessment, Shanghai Ocean University, Shanghai 201306, China

**Keywords:** mitogenome, high-throughput sequencing, *Diogenes edwardsii*, Anomura, phylogenetic analyses

## Abstract

A complete mitochondrial genome (mitogenome) can provide important information for gene rearrangement, molecular evolution and phylogenetic analysis. Currently, only a few mitogenomes of hermit crabs (superfamily Paguridae) in the infraorder Anomura have been reported. This study reports the first complete mitogenome of the hermit crab *Diogenes edwardsii* assembled using high-throughput sequencing. The mitogenome of *Diogenes edwardsii* is 19,858 bp in length and comprises 13 protein-coding genes, 2 ribosomal RNA genes, and 22 transfer RNA genes. There are 28 and six genes observed on the heavy and light strands, respectively. The genome composition was highly A + T biased (72.16%), and exhibited a negative AT-skew (−0.110) and positive GC-skew (0.233). Phylogenetic analyses based on the nucleotide dataset of 16 Anomura species indicated that *D. edwardsii* was closest related to *Clibanarius infraspinatus* in the same family, Diogenidae. Positive selection analysis showed that two residues located in *cox1* and *cox2* were identified as positively selected sites with high BEB value (>95%), indicating that these two genes are under positive selection pressure. This is the first complete mitogenome of the genus *Diogenes*, and this finding helps us to represent a new genomic resource for hermit crab species and provide data for further evolutionary status of Diogenidae in Anomura.

## 1. Introduction

In the past twenty years, the mitochondrial genome (mitogenome) has been widely used in the studies of molecular evolution and reconstruction of phylogeny [1,2]. Nucleotide sequences or amino acid data, gene arrangement and RNA secondary structures of mitogenomes are greatly useful for exploring the relationships between metazoan lineages [2,3]. Generally, a metazoan mitogenome is a closed circular DNA molecule, ranging in size from 14 to 20 kb, and typically contains 37 genes: 13 protein-coding genes (PCGs) (*cob*, *cox1-3*, *atp6*, *atp8*, *nad1-6*, and *nad4l*), 2 rRNA genes (*rrnS* and *rrnL*), and 22 tRNA genes [4]. In addition, there are always several noncoding regions in the mitogenome, and the longest noncoding “AT-rich” region is known as the control region, which contains some of the initiation sites for genome transcription and replication [4]. Mitochondrial DNA generally forms an independent unit of genetic information that evolves independently of the nuclear genome [5]. Due to their haploid characteristics, maternal inheritance, limited recombination and rapid rate of evolution, mitogenomes have been commonly used for evolutionary and phylogenetic studies [6]. With the rapid development of sequencing technology, next-generation sequencing has become a fast and low-cost method to provide complete mitogenomes [7].

The infraorder Anomura is a collection of diverse crustaceans placed within the Decapoda with over 2500 species [8]. Anomura species have an exceptional range of body forms and can be found in diverse habitats including hydrothermal vents, freshwater bodies, and terrestrial environments [9]. The latest classification scheme divides Anomura into seven superfamilies: Aegloidea, Chirostyloidea, Galatheoidea, Hippoidea, Lithodoidea, Lomisoidea, and Paguroidea. Hermit crabs (the superfamily Paguroidea Latreille, 1802) are one of the best known and most diverse consisting of Coenobitidae, Diogenidae, Paguridae, Parapaguridae, Pylochelidae, and Pylojacquesidae, with more than 1100 species inhabiting diverse biotopes from intertidal to deep seas [10]. The hermit crabs represent an intermediate group of crustaceans from Macrura to Brachyura, which occupy a key position in crustacean evolution [11]. These species are unique because they have asymmetrical bodies that allow them to live in gastropod shells. However, merely a few of the evolutionary relationships of hermit crabs at taxonomic levels have been investigated, which are still waiting for researchers to further resolve [12,13,14]. 

Diogenidae Ortmann, 1892 is the second largest family of the superfamily Paguroidea, including 22 genera and 482 species worldwide (http://www.marinespecies.org/, accessed on 28 September 2022). Within Diogenidae, hermit crabs of the diogenid genus *Diogenes* Dana, 1851 contains about 60 species [15,16]. Most studies of species in the genus *Diogenes* were described based on their morphology [15,17,18], and the mitogenomes of the species in genus *Diogenes* have never been reported, nor have their phylogenies within the infraorder Anomura. The species *Diogenes edwardsii* was considered to constitute a scavenging force as important as nassariid gastropods, showing an important ecological value in the marine ecosystem [19]. Up to now, only some morphological features and feeding behavior information of *D. edwardsii* have been investigated [19]. As reported, *D. edwardsii* is characteristic in the spinose left chela, lacking a distinct crest on the outer surface, having possession of one prominent spine on the ventromesial margin proximally on the merus of the left cheliped, and the propodus of the left third pereopod being armed with additional rows of spinules on the lateral surface [19].

Prior to this study, the mitogenomic characteristics of *D. edwardsii* have never been reported. We newly sequenced and annotated the first complete mitogenome of the hermit crab *D. edwardsii* based on specimen from the Bohai Bay in China. The mitogenome organization and codon usage of *D. edwardsii* were revealed. The phylogenetic relationships of Anomura were reconstructed based on published mitogenomes to explore the potential status of *D. edwardsii*. We also performed positive selection analysis of Anomura mitochondrial PCGs to understand the adaptive evolution of *D. edwardsii* in Anomura.

## 2. Materials and Methods

### 2.1. Sample Collection and DNA Extraction

An adult female *D. edwardsii* was collected form Bohai Bay (38°32.12′ N, 118°1.50′ E), China, in November 2021. Species-level morphological identification was carried out according to the main ideas of Komai et al. [18]. The specimen was preserved in 95% ethanol and stored at −80 °C in dark. Genomic DNA was extracted from the specimen’s muscle tissue using the SDS method [20]. The harvested DNA was detected using agarose gel electrophoresis and quantified by a Qubit^®^ 2.0 Fluorometer (Thermo Scientific, Shanghai, China). The DNA and voucher specimen of *D. edwardsii* was deposited in the College of Environmental Science and Engineering at Nankai University (Tianjin, China).

### 2.2. Mitogenome Sequencing and Assembly

After DNA isolation, 1 μg of purified DNA was fragmented. Paired-end libraries (450 bp) were prepared following the manufacturer’s instructions, then sequenced on the Illumina NovaSeq 6000 at Shanghai BIOZERON Co., Ltd. (Shanghai, China).

Raw reads were filtered into clean reads prior to assembly, and then undesirable reads were removed. These were reads with adaptors; low quality reads (i.e., those showing a quality score below 20 (Q < 20)); reads which comprised more than 10% of unknown bases (N); and any duplicated sequences. The mitochondrial genome was reconstructed using a combination of de novo and reference-guided assemblies, and the following three steps were performed to assemble the mitogenome. Firstly, the filtered reads were assembled into contigs using SPAdes 3.14.1 (http://bioinf.spbau.ru/spades, accessed on 24 March 2022). Then, BLAST (https://blast.ncbi.nlm.nih.gov/Blast.cgi, accessed on 24 March 2022) was used to align contigs with reference mitogenomes from Anomura species, and aligned contigs (≥80% similarity and query coverage) were ordered according to the reference mitogenomes. Finally, GapCloser 1.12 with default parameters were used to map the clean reads to the assembled mitogenome, and most gaps were filled with local assembly.

The mitochondrial genes were annotated using the online MITOS tool (http://mitos.bioinf.uni-leipzig.de/index.py, accessed on 24 March 2022), and the protein-coding genes, transfer RNA (tRNA) genes, and ribosome RNA (rRNA) genes were predicted with default parameters. BLAST searches for reference mitochondrion genes were used to determine the location of each coding gene. Manual corrections of genes for start/stop codons were performed in SnapGene Viewer by referring to the reference mitogenomes. The circular map of mitogenome of *D. edwardsii* was drawn by the CGview tool (http://stothard.afns.ualberta.ca/cgview_server/, accessed on 24 March 2022).

### 2.3. Sequence Analysis

The nucleotide composition and codon usage were computed using DnaSP 5.1 [21]. The AT and GC skews were calculated with the following formulas: AT skew = (A − T)/(A + T) and GC skew = (G − C)/(G + C) [22], where A, T, G, and C are the occurrences of the four nucleotides. The complete mitochondrial DNA sequence was deposited in GenBank with the accession number OP047688.

### 2.4. Phylogenetic Analysis

Along with the complete mitogenome DNA sequence from *D. edwardsii*, all currently complete mitochondrial sequences of 16 Anomura species were used in phylogenetic analysis (Table 1). *Ovalipes punctatus* (GenBank: NC_042695) and *Callinectes sapidus* (GenBank: NC_006281) were used as outgroups (Table 1). Our data set was based on nucleotide sequences from 13 mitochondrial PCGs (*cox1*, *cox2*, *cox3*, *cob*, *atp6*, *atp8*, *nad1*, *nad2*, *nad3*, *nad4*, *nad4l*, *nad5*, and *nad6*). Multiple alignments of the whole 13 genes were conducted using MUSCLE 3.8.31. Ambiguously aligned regions and gaps in aligned sequences were removed using Gblocks server 0.91b [21] with the default setting. ModelTest 2.1.10 [22] was used to select the best-fit evolutionary models GTR + I + G for the nucleotide dataset. The Maximum likelihood (ML) analysis of phylogenetic trees and model parameters were performed by RAxML 8.2.8 [23], and the best model was determined through AIC and BIC scores. Topological robustness for the ML analysis was evaluated with 100 replicates of bootstrap support. The phylogenetic tree (http://www.atgc-montpellier.fr/phyml/, accessed on 22 July 2022) was constructed by PhyML v3.0 and graphically edited with the iTOL 3.4.3 (https://itol.embl.de/itol.cgi, accessed on 22 July 2022).

### 2.5. Positive Selection Analysis

As reported in [24], comparing the nonsynonymous/synonymous substitution ratios (ω = dN/dS) has become a useful approach for quantifying the effect of natural selection on molecular evolution: ω > 1, =1 and <1 represent positive selection, neutrality and purifying selection, respectively [25]. The codon-based maximum likelihood (CodeML) method implemented in the PAML package [26] was applied to estimate the dN/dS ratio ω. The combined database of 13 mitochondrial PCGs was used for the selection pressure analyses. The ML phylogenetic tree was used as the working topology in the CodeML analyses.

We used branch models in the present study to evaluate positive selection in the infraorder Anomura. First, under an assumption of no adaptive evolution in the gene sequences, the one-ratio model (M0) estimated the distribution of ω values as a benchmark, which assumes a single ω ratio for all branches in the phylogenetic relationships [27]. Then, the two-ratio model (M2) that allows the background lineages and foreground lineages to have distinctive ω ratio values was used to detect positive selection acting on branches of interest [28,29]. Last, a free-ratio model (M1), which allows ω ratio variation among branches, was used to estimate ω values on each branch [29]. The one-ratio model (M0) and the free-ratio model (M1) were compared to determine whether different clades in Anomura had different ω values. In addition, the one-ratio model (M0) and the two-ratio model (M2) were used to compare whether Paguroidea species underwent more selection pressure than other Anomura species. Here, ω0 and ω1 represent the values for the background branch and foreground branch, respectively. Pairwise models were compared with the critical values of the Chi square (χ^2^) distribution using likelihood ratio tests (LRTs). The test statistic was estimated as twice the difference in log likelihood (2ΔL) and the degrees of freedom were estimated as the difference in the number of parameters for each model.

Additionally, branch-site models were fitted to examine the positive selection on some sites among the Anomura species. Branch-site models allow ω to vary both among sites in the protein and across branches on the tree. Branch-site model A (positive selection model) was used to identify the positive selected sites among the lineages of Paguroidea species (marked as foreground lineages). The sites with ω > 1 suggest that model A fits the data significantly better than the corresponding null model. Bayes Empirical Bayes analysis was used to calculate posterior probabilities to identify positive selection sites on the foreground lineages if the LRTs was significant [30].

## 3. Results and Discussion

### 3.1. Dedwardsii Mitogenome Organization

The Illumina HiSeq runs resulted in 59,106,998 paired-end reads from the *D. edwardsii* library with an insert size of approximately 450 bp. Selective-assembly analysis showed that 185,891 reads were assembled into a 19,858 bp circular molecule, representing the complete mitogenome of *D. edwardsii* (Figure 1 and Table 2). This length is longer than the mitogenome length of other reported Anomura species, which ranges from 15,324 bp in *N. maculatus* to 17,910 bp in *Munidaisos* (Table 1). The genome encodes 37 genes, including 13 PCGs, 2 rRNA genes, and 22 tRNA genes. A total of 28 genes are encoded on the heavy (H) strand, and nine genes are encoded on the light (L) strand (Table 2). The intergenic regions are distributed in 28 locations, and the longest region is 1971 bp (between *trnS2-tca* and *nad1*). Meanwhile, 31 overlapping nucleotides are located in four pairs of neighboring genes for the mitogenome. These overlapping nucleotides vary in length from 1 to 18 bp, and the longest overlap is located between *rrnL* and *trnV-gta*. The *D. edwardsii* mitogenome has a nucleotide composition of 32.11% A, 10.68% C, 17.16% G, and 40.05% T and an overall AT content of 72.16%. For the *D. edwardsii* mitogenome, the AT skew is −0.110, and the GC skew is 0.233, which indicates bias toward T and G. This asymmetry often occurs on Anomura species [11,26,31,32]. In mammals, the duration of single-stranded state of the “heavy-stranded” genes during mitochondrial DNA replication can explain this asymmetry [32]. Whether or not the same explanation works on our results remains difficult to predict at this time due to the scarcity of information regarding DNA replication of invertebrate mitochondria.

For most families of the order Decapoda, congeners belonging to the same family generally possess identical gene arrangement [11], which has particularly been confirmed based on the reported crab mitogenomes [26,27,28]. However, comparing with the reported mitogenomes of the species *C. infraspinatus* and *D. arrosor* belonging to the same family Diogenidae, the *D. edwardsii* mitogenome possesses a distinctive gene arrangement [11,29]. For instance, *nad5* is located between *trnH-cac* and *trnF-ttc* in the mitogenome of *D. edwardsii*, instead of the common location between *trnF-phe* and *trnH-his* in the mitogenomes of *C. infraspinatus* and *D. arrosor*. The gene rearrangement found in the *D. edwardsii* mitogenome deepens our understanding of this genomic feature about hermit crabs, and represents a special gene order in the genus *Diogenes*.

### 3.2. Protein-Coding Genes

The total length of all 13 PCGs of *D. edwardsii* is 10,888 bp, accounting for 54.83% of the complete length of the mitogenome (Table 2). The 13 PCGs ranged in size from 159 bp (*atp8*) to 1707 bp (*nad5*), and all 13 PCGs are encoded on the heavy strand (Table 2). Furthermore, there are two reading-frame overlaps on the same strand: *atp6* and *atp8*, and *atp6* and *cox3* each share seven and one nucleotides (Table 2). The PCGs encode 3629 amino acids, among which *Ser* (16.19%) and *Leu* (9.84%) are the most and frequently used, and *Met* (1.64%) and *Trp* (1.64%) are the least frequently used, respectively. The A + T contents of PCGs and third codon positions in *D. edwardsii* mitogenome are 69.08% and 72.30%, respectively. The strong AT-bias in the third codon positions is similar to many species in the infraorder Anomura, i.e., *D. arrosor*, *Shinkaia crosnieri* and *P. longicarpus* [11,31,32].

Codon usage bias is a phenomenon in which specific codons are used more frequently than other synonymous codons by certain organisms during the translation of genes to proteins [11]. As shown in Figure 2, a total of 61 available codons are used in *D. edwardsii* mitogenome. Start and stop codons varied among the PCGs. Four of 13 PCGs use ATT (*cox1*, *atp8*, *nad6*, and *nad1*) and ATA (*nad2*, *cob*, *nad4*, and *nad5*) as start codons, and three PCGs use ATG (*cox2*, *atp6*, and *cox3*) and *nad3* uses GTG. Seven of 13 PCGs use TAA (*cox1*, *cox2*, *atp6*, *cox3*, *cob*, *nad1*, and *nad4l*) as a stop codon, and five PCGs use TAG (*nad2*, *atp8, nad4*, *nad5*, and *nad3*). Finally, *nad6* exhibits an incomplete (A) stop codon. The presence of an incomplete stop codon is a common phenomenon in both invertebrate and vertebrate mitochondrial genes [30,33,34]. It is assumed that truncated stop codons are completed via post-transcriptional polyadenylation [35]. Previous studies have shown that truncated stop codons were common in the metazoan mitogenomes and may be corrected by post-transcriptional polyadenylation [36,37]. The relative synonymous codon usage (RSCU) values indicated that the six most commonly used codons are TTA(*Leu*), TCT(*Ser*), CCT(*Pro*), ACT(*Thr*), GCT(*Ala*), and AGA(*Ser*) (Figure 2). Based on RSCU and amino acid composition in the PCGs, comparative analyses showed that the acid composition kept similar with most hermit crabs, whereas the codon usage pattern of *D. Edwardsii* is unidentified with some Paguroidea species. The codon TCT(*Ser*) instead of TTA(*Leu*) was often found to be used most frequently in Paguroidea species mitogenomes, i.e., *D. arrosor*, *B. latro*, *P. longicarpus*, and *Coenobita rugpsus* [9,11,32,38], while without the most frequency in *D. edwardsii*.

### 3.3. Ribosomal RNA and Transfer RNA Genes

The *rrnL* and *rrnS* genes of *D. edwardsii* are 1370 bp (AT% = 67.2) and 796 bp (AT% = 67.7) in length, respectively. The total length of rRNA genes accounts for 10.09% of the *D. edwardsii* mitogenome. Both of the two genes are encoded on the light strand. The AT and GC content of rRNAs are 77.42% and 22.58%, and the AT skew and GC skew are −0.008 and 0.010, respectively, suggesting a slight bias toward the use of T and G.

A total of 22 tRNA genes were identified in the mitogenome of *D. edwardsii* (Table 2). The length of tRNA genes is 1491 bp, accounting for 7.51% of the complete length of the mitogenome. A total of 15 genes (*trnL1-cta*, *trnK-aaa*, *trnD-gac*, *trnR-cga*, *trnN-aac*, *trnE-gaa*, *trnT-aca*, *trnS2-tca*, *trnP-cca*, *trnH-cac*, *trnF-ttc*, *trnC-tgc*, *trnL2-tta*, *trnG-gga*, *trnS1-aga*) are encoded on the heavy strand and the rest four genes (*trnL-atc*, *trnM-atg*, *trnY-tac*, *trnV-gta*) are encoded on the light strand. The AT skew of tRNA genes is 0.013, and the GC skew is 0.108, indicating a bias toward A and G. In addition, the codons with A and T in the third position are used more frequently than other synonymous codons. These characteristics reflect codon usage with A and T bias at the third codon position, similar to the bias found in many metazoans [39,40,41].

### 3.4. Phylogenetic Relationships

In the present study, the phylogenetic relationship of *D. edwardsii* was constructed on the sequences of the concentrated 13 PCGs from the mitochondrial genomes of 16 Anomura species and two outgroups (Figure 3). Phylogenetic analyses were carried out based on nucleotide sequences of 13 PCGs by the ML method. Anomura was divided into two main branches in our phylogenetic tree, which was identified in the previous researches [24,42]. The first clade comprised five families (Porcellanidae, Munididae, Paguridae, Lithodidae, and Kiwaidae) and the second clade contained two families (Coenobitidae and Diogenidae). It is not difficult to find that the two families of the superfamily Paguroidea belong to two branches of the phylogenetic tree, which presents challenges for the current classification based on the morphology. Even if inter-superfamiliar relationships are not fully resolved, clear trends become visible. The phylogenetic relationships of these seven families show as being ((((Porcellanidae + Munididae) + Paguridae) + Lithodidae) + Kiwaidae) + (Coenobitidae + Diogenidae).

Consistent with the traditional morphological classification, the phylogenetic analyses based on molecular methods showed that *D. edwardsii* clustered with the Diogenidae species *C. infraspinatus* and *D. arrosor*. However, the members of the family Coenobitidae *B. latro* and *C. brevimanus* are clustered within the same clade. As reported, *C. infraspinatus* possesses a favor of codon TAT(*Ser*) and owns the highest *Ser* content in amino acid composition [29], which showed the same pattern with *D. edwardsii*. This supports the phylogenetic analysis result that these two species possess a close relationship. The ML bootstrap values supporting *D. edwardsii* and *C. infraspinatus* as well as *D. arrosor* were not high (<80), which may be explained by the different gene arrangements between *D. edwardsii* and these two species, as discussed in Section 3.1. Thus, more species mitogenomes in the family Diogenidae still need to be investigated to fulfill the systematic phylogeny within the infraorder Anomura. Our newly acquired mitogenome data and phylogenetic results can also be better used to provide a basis for studies of the mitochondrial evolution of Anomura.

### 3.5. Positive Selection Analysis

The ω(dN/dS) ratio calculated in branching model analysis *D. edwardsii*’s 13 mitochondrial PCGs was 0.02684 under the one-ratio model (M0) (Table 3), indicating that these genes undergo constrained selection pressure to maintain primary function. When using two-ratio model, the ω ratio of the *D. edwardsii* branch possess no significant difference compared with that in another branch (ω1 = 0.02703 and ω0 = 0.02683), indicating similarity in the selection pressure between the species in the family Coenobitidae and Diogenidae and other Anomura species. However, in the comparison of the one-ratio model (M0) and the free-ratio model (M1), the LRTs indicated that the free-ratio model fit the data better than the one-ratio model (Table 3), representing that the ω ratios are distinctive among different lineages.

Positive selection often occurs over a short evolutionary time and performs on only a few sites [43,44]. Thus, the signals of positive selection are often overwhelmed by those for successive purifying selection that occur at most sites in the gene sequence [43,44,45]. In the current study, branch-site models were used to determine possible positively selected sites in *D. edwardsii* (Table 4). Two residues, located in *cox1* and *cox2*, were identified as positive selected sites in *D. edwardsii* (>95%), respectively, indicating that these two genes are under positive selection pressure. It is known that mitochondrial PCGs are crucial in the oxidative phosphorylation pathway. Cytochrome *c* oxidase catalyzes the oxygen terminal reduction, and the catalytic core is encoded by three mitochondrial PCGs (*cox1*, *cox2*, and *cox3*) [45,46]. Cytochrome c oxidase is confirmed to be a particularly important site of the positive selection in marine anoxic adaptation [47,48]. This study also indicates the mechanism applicability of Cytochrome c oxidase on the hermit crabs living in the specific marine environment.

The cytochrome c oxidase requires reactive oxygen species (ROS) when the living cells are exposed to anoxia. The increase in ROS concentration helps to stabilize Hif-1α, which then results in the induction of Hif-1-dependent nuclear hypoxic genes [48,49,50]. In this study, the positive selection sites in *cox1* and *cox2* of *D. edwardsii* indicate a phylogenetic adaption of ATP synthase of hermit crabs in the presence of reduced oxygen availability or increased energy requirements. *Cox1* and *cox2* in mitochondrion are involved in the regulation of platelet aggregation and vasomotor to maintain the stability of physiological functions of cells and tissues [45]. Functional modification mediated by positive selection pressure in *D. edwardsii* may increase the affinity between the enzyme and oxygen, and then efficiently maintain basic metabolic levels under a hypoxia environment in the marine [45]. The marine environment is characterized by a lack of high pressure and low dissolved oxygen, both of which could directly affect the mitochondrial aerobic respiration process [51]. The positive selected sites found in *cox1* and *cox2* may be associated with the adaptive evolution of *D. edwardsii* within the hermit crabs living in the marine environment. Our study provides a foundation for understanding the mitochondrial evolution mechanisms of *D. edwardsii* to obtain adequate energy and maintain essential metabolic levels in specific marine ecosystem.

## 4. Conclusions

In this study, we determined and described the complete mitogenome of *D. edwardsii*, supplementing the first mitogenome information of the genus *Diogenes* (Anomura: Paguroidea: Diogenidae). Our phylogenetic tree based on molecular methods showed that *D. edwardsii* possessed the closest relationship with the species *C. infraspinatus* in the same family. Positive selection analysis showed that *cox1* and *cox2* of *D. edwardsii* were under positive selection pressure, indicating the potential mechanisms of this species that efficiently maintain basic metabolic levels under a hypoxia environment in the marine ecosystem. Our results provide useful mitogenome information for a better understanding of the hermit crabs, as well as the phylogenetics of *D. edwardsii* in the infraorder Anomura.

## Figures and Tables

**Figure 1 genes-14-00470-f001:**
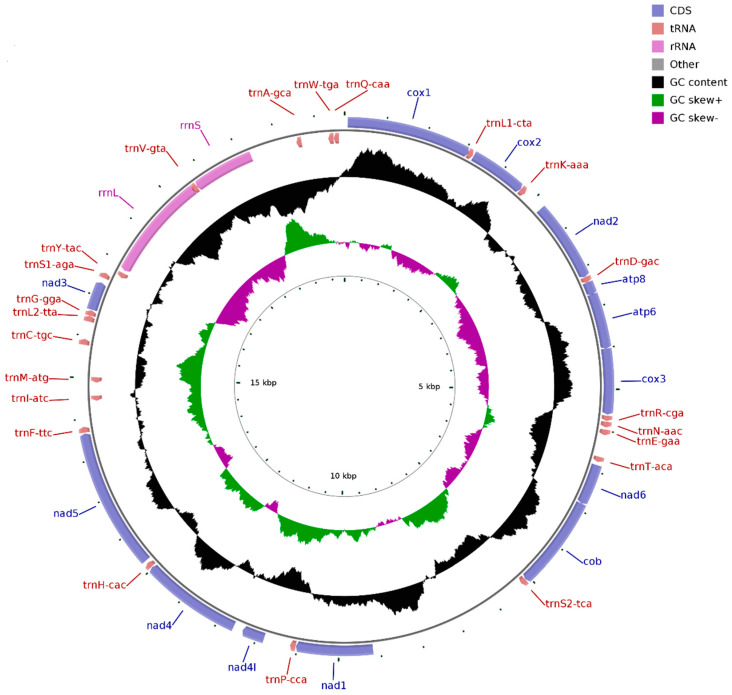
Complete mitogenome map of *D. edwardsii*. A total of 28 genes are encoded on the heavy (H) strand and nine genes are encoded on the light (L) strand. Genes for proteins and rRNAs are represented with standard abbreviations. Genes for tRNAs are shown in a single letter for the corresponding amino acid, with two leucine tRNAs and two serine tRNAs distinguished by numerals.

**Figure 2 genes-14-00470-f002:**
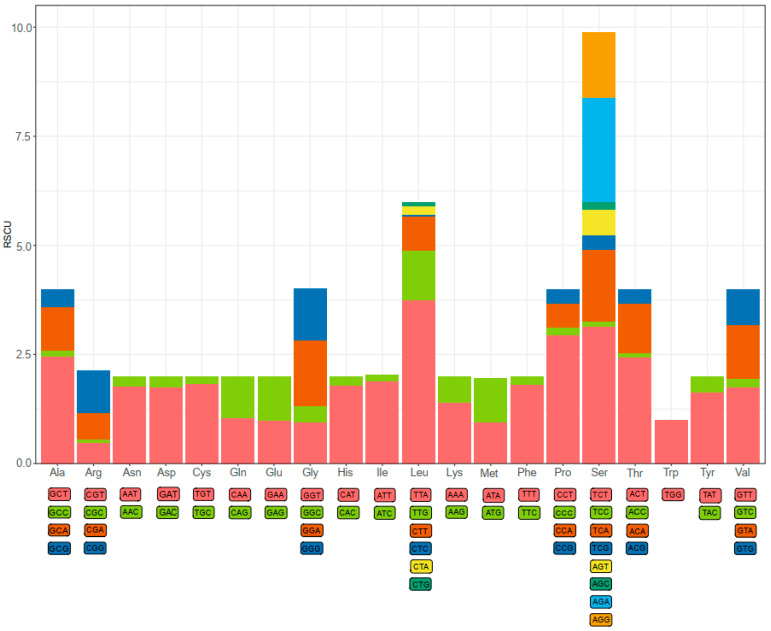
The relative synonymous codon usage (RSCU) of the *D. edwardsii* mitogenome.

**Figure 3 genes-14-00470-f003:**
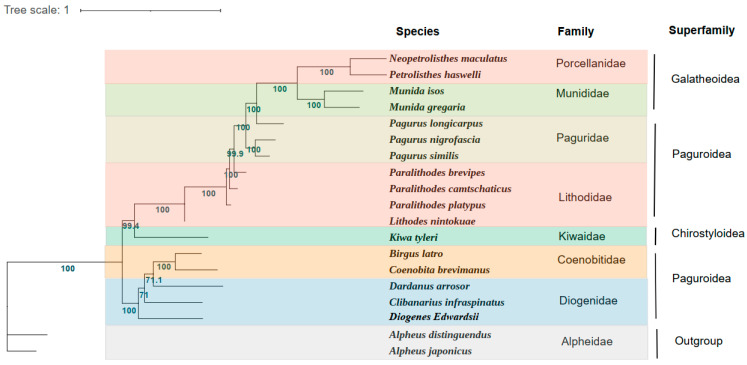
Phylogenetic tree derived from Maximum likelihood (ML) analysis based on 13 mitochondrial PCGs (*atp6*, *atp8*, *cox1*, *cox2*, *cox3*, *cob*, *nad1*, *nad2*, *nad3*, *nad4*, *nad4L*, *nad5*, and *nad6*). Species belonging to the same superfamily are marked with the same color. Numbers on branches are Bayesian posterior probabilities (percent).

**Table 1 genes-14-00470-t001:** List of Anomura species with their GenBank accession numbers.

Superfamily	Family	Species	Size(bp)	Accession. no
Chirostyloidea	Kiwaidae	*Kiwa tyleri*	16,865	NC_034927
Galatheoidea	Munididae	*Munida gregaria*	16,326	NC_030255
		*Munida isos*	17,910	NC_039112
	Porcellanidae	*Neopetrolisthes maculatus*	15,324	NC_020024
		*Petrolisthes haswelli*	15,348	NC_025572
Paguroidea	Coenobitidae	*Birgus latro*	16,411	NC_045091
		*Coenobita brevimanus*	16,393	NC_050386
	Diogenidae	*Clibanarius infraspinatus*	16,504	NC_025776
		*Dardanus arrosor*	16,592	NC_060631
		*Diogenes edwardsii*	19,858	OP 047688
	Lithodidae	*Lithodes nintokuae*	15,731	NC_024202
		*Paralithodes brevipes*	16,303	NC_021458
		*Paralithodes camtschaticus*	16,720	NC_020029
		*Paralithodes platypus*	16,883	NC_042240
	Paguridae	*Pagurus longicarpus*	15,630	NC_003058
		*Pagurus nigrofascia*	15,423	NC_042412
		*Pagurus similis*	17,100	NC_057304
Outgroup	Alpheidae	*Alpheus distinguendus*	15,700	NC_014883
		*Alpheus japonicus*	16,619	NC_038116

**Table 2 genes-14-00470-t002:** Mitogenome organization of *D. edwardsii*.

Gene	Strand	Position	Length	Intergenic Region	Start Coden	Stop Coden	Anticodon
*cox1*	H	37–1539	1503	-	ATT	TAA	
*trnL1-cta*	H	1535–1601	67	−5	-	-	CUA
*cox2*	H	1613–2302	690	11	ATG	TAA	
*trnK-aaa*	H	2305–2374	70	2	-	-	AAA
*nad2*	H	2639–3613	975	264	ATA	TAG	
*trnD-gac*	H	3617–3685	69	3	-	-	GAC
*atp8*	H	3686–3844	159	0	ATT	TAG	
*atp6*	H	3838–4512	675	−7	ATG	TAA	
*cox3*	H	4512–5303	792	−1	ATG	TAA	
*trnR-cga*	H	5318–5381	64	14	-	-	CGA
*trnN-aac*	H	5391–5459	69	9	-	-	AAC
*trnE-gaa*	H	5486–5554	69	26	-	-	GAA
*trnT-aca*	H	5820–5884	65	265	-	-	ACA
*nad6*	H	5923–>6418	496	38	ATT	A	
*cob*	H	6419–7549	1131	0	ATA	TAA	
*trnS2-tca*	H	7553–7619	67	3	-	-	UCA
*nad1*	H	9591–10,517	927	1971	ATT	TAA	
*trnP-cca*	H	10,521–10,588	68	3	-	-	CCA
*nad4l*	H	10,911–11,183	273	322	GTG	TAA	
*nad4*	H	11,300–12,514	1215	116	ATA	TAG	
*trnH-cac*	H	12,521–12,586	66	6	-	-	CAC
*nad5*	H	12,620–14,326	1707	33	ATA	TAG	
*trnF-ttc*	H	14,337–14,407	71	10	-	-	UUC
*trnI-atc*	L	14,711–14,776	66	303	-	-	AUC
*trnM-atg*	L	14,941–15,009	69	164	-	-	AUG
*trnC-tgc*	H	15,395–15,465	71	385	-	-	UGC
*trnL2-tta*	H	15,678–15,748	71	212	-	-	UUA
*trnG-gga*	H	15,750–15,815	66	1	-	-	GGA
*nad3*	H	15,825–16,169	345	9	GTG	TAG	
*trnS1-aga*	H	16,230–16,295	66	60	-	-	AGA
*trnY-tac*	L	16,322–16,387	66	26	-	-	UAC
*rrnL*	L	16,423–17,792	1370	35	-	-	
*trnV-gta*	L	17,775–17,843	69	−18	-	-	GUA
*rrnS*	L	17,844–18,639	796	0	-	-	
*trnA-gca*	L	19,241–19,306	66	601	-	-	GCA
*trnW-tga*	L	19,649–19,717	69	342	-	-	UGA
*trnQ-caa*	L	19,718–19,784	67	0	-	-	CAA

**Table 3 genes-14-00470-t003:** CodeML analyses of selection pressure on mitochondrial genes in Paguroidea lineage.

Trees	Models	lnL	Parameter Estimates	Model Compared	2ΔL	LRT *p*-Value
Branch models						
ML tree	M0	−120,734.213411	ω = 0.02684			
	Free-ratio	−119,787.8256		Free-ratio vs. M0	1892.77562	0.00000
	Two-ratio	−120,734.211446	ω0 = 0.02683 ω1 = 0.02703	Two-ratio vs. M0	0.00393	0.95001
Branch-sits models			p0 = 0.90771; p1 = 0.03533; p2a = 0.05483; p2b = 0.00213			
ML tree	Null model	−119,878.534338	ω0 =0.02575; ω1 = 1.00000; ω2a = 1.00000; ω2b = 1.00000			
	Model A	−119,874.014382	p0 = 0.91806; p1 = 0.03578; p2a = 0.04443; p2b = 0.00173 ω0 =0.02593; ω1 = 1.00000; ω2a = 4.09193; ω2b = 4.09193	Model A vs. null model	9.03991200000746	0.002641484

**Table 4 genes-14-00470-t004:** Possible sites under positive selection in Paguroidea lineage.

ML Tree			
Gene	Codon	Amino Acid	BEB Values
cox1	1839	M	0.981 *
cox2	2810	S	0.958 *

* 0.95 < BEB < 0.99.

## Data Availability

The mitochondrial genome has been deposited in the NCBI with accession number OP047688.

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
