# Peer review of "The Complete Mitochondrial Genome of the Hermit Crab Diogenes edwardsii (Anomura: Diogenidae) and Phylogenetic Relationships within Infraorder Anomura"

_genes, 2023, doi:10.3390/genes14020470_

Round 1

Reviewer 1 Report

Only in the reference section authors have to present the reference in uniform style. In the result section some typographical mistakes are present. These should be rectified.

Please take necessary steps to change the species name to small letter instead of capital letter: Example: Diogenes edwardsii  instead of Diogenes Edwardsii.

Author Response

Only in the reference section authors have to present the reference in uniform style. In the result section some typographical mistakes are present. These should be rectified.

Please take necessary steps to change the species name to small letter instead of capital letter: Example: Diogenes edwardsii instead of Diogenes Edwardsii.

>> The reference section has been carefully checked and revised. All of the capital letters in the species names (Diogenes Edwardsii) have been changed into small letters (Diogenes edwardsii).

Reviewer 2 Report

The manuscript presents the first report on whole mt-Genome sequencing and annotation of Diogenes edwardsii. Overall, the data is well presented and can be published if the authors improve the quality of the figures and address the comparative analysis on how good the sequence and sequence annotation quality is in comparison with the existing literature. Authors can also address how this genomic data will be useful for conservation purposes.

Overall the language and typographical errors can be rectified in the revised version.

Author Response

The manuscript presents the first report on whole mt-Genome sequencing and annotation of Diogenes edwardsii. Overall, the data is well presented and can be published if the authors improve the quality of the figures and address the comparative analysis on how good the sequence and sequence annotation quality is in comparison with the existing literature. Authors can also address how this genomic data will be useful for conservation purposes.

Overall the language and typographical errors can be rectified in the revised version.

>> Many thanks for the constructive comments. The quality of the figures has been improved. And in lines 239-248, we have compared the existing literature with the current study, to prove the quality of the sequence annotation. The language of the manuscript has been improved in this version.

Reviewer 3 Report

Please check for the scientific name of the research object according to Zoological Code. The quality of the figures must ne improved.

Author Response

Please check for the scientific name of the research object according to Zoological Code. The quality of the figures must be improved.

>> The scientific name of the research object has been checked according to Zoological Code. The quality of the figures has been improved.

Reviewer 4 Report

This is a clear study. I have only minor comments and some suggestions:

Diogenes Edwardsii ← is this correct or shouldn’t it be Diogenes edwardsii

Line 46-47 “Mitochondrial DNA forms an independent unit of genetic information that evolves independently of the nuclear genome [5]” – this may not always be the case e.g. when female are heterogametic as in birds ZW the mitogenome is transferred with W and are thus a single linkage block and does not evolve independently from the W. There may also be some interactions between nuclear genes and mitochondria

Line 48 are increasingly used ← have been commonly used

Line 48 mitotic genomes ← mitogenomes 

Line 70 were reported about their morphology ← rephrase, e.g.were secribed based on their morphology

line 74 behiviour← behaviour

Figure 1 GC content is both above and below the circle – add description in legend how to read the figure 

Figure 1 legend  “..” ← “.”

Line 210 For instance, nad5 located ← add “is”  as : For instance, nad5 [is] located

Line 213-214 “As a result, the subtle differences in gene arrangement between these species will present new challenges to this conclusion.” - delete or rephrase better“ will present new challenges to this conclusion” this is a finding/result not a conclusion

line 287-288  “was clustered firstly clustered with the Diogenidae species Clibanarius infraspinatus and then clustered with another Diogenidae species Dardanus arrosor.” ← rephrase “clustered with the Diogenidae species Clibanarius infraspinatus and  Dardanus arrosor. However, the members of the family Coenobitidae … clustered within the same clade.” 

line 292 “The ML bootstrap values supporting D. Edwardsii and Clibanarius infraspinatus as well as Dardanus arrosor were not high (<80), which may be explained by the different gene arrangements between D. Edwardsii and these two species, as discussed in part 3.1.” check his is not correct. The overall support for this clade is 100 but the branches or ordering of  Clibanarius infraspinatus and  Dardanus arrosor within the clade is not well supported, i.e. whether C. infraspinatus or D. arrosor is more related to D. Edwadsii than the other.

Line 294-295 “Thus, more species mitogenomes in the family Diogenidae still need to be investigated to fulfill the systematic phylogeny within the infraorder Anomura” – yes or nuclear markers. Even though better support would be achieved this is just the phylogeny of the mitogenomes which can differ from the nuclear genome and thus nuclear markes are also needed to recontruct  the phylogeny of the species.

Lines 298-302 “Comparing the nonsynonymous/synonymous substitution ratios (ω = dN/dS) has be299 come a useful approach for quantifying the impact of natural selection on molecular evo300 lution [26]. ω >1, = 1 and <1 indicate positive selection, neutrality and purifying selection, 301 respectively [52]. We examined potential positive selection in the Paguroidea lineage 302 (family Coenobitidae and Diogenidae in Fig. 3) using CodeML from the PAML package. “ true  this is in Methods and can be deleted here. 

“no significant significance” – rephrase

Conclusion-  lines 350-351: “Our phylogenetic tree showed no controversies with the traditional morphological classification based on molecular methods that D. Edwardsii possessed the closest relationship with the species Clibanarius infraspinatus in the same family.” But  Coenobitidae clustered within the  Diogenidae family, suggesting that this is a single family.

Author Response

This is a clear study. I have only minor comments and some suggestions:

Diogenes Edwardsii ← is this correct or shouldn’t it be Diogenes edwardsii

>> All of the capital letters in the species names (Diogenes Edwardsii) have been changed into small letters (Diogenes edwardsii).

Line 46-47 “Mitochondrial DNA forms an independent unit of genetic information that evolves independently of the nuclear genome [5]” – this may not always be the case e.g. when female are heterogametic as in birds ZW the mitogenome is transferred with W and are thus a single linkage block and does not evolve independently from the W. There may also be some interactions between nuclear genes and mitochondria.

>> Thanks for your correction. The sentence has been revised as follows: Mitochondrial DNA generally forms an independent unit of genetic information that evolves independently of the nuclear genome.

Line 48 are increasingly used ← have been commonly used

>> Revised.

Line 48 mitotic genomes ← mitogenomes

>> Revised.

Line 70 were reported about their morphology ← rephrase, e.g.were secribed based on their morphology

>> The sentence has been revised as suggested.

line 74 behiviour← behaviour

>> Revised.

Figure 1 GC content is both above and below the circle – add description in legend how to read the figure

>> Description in legend has been added.

Figure 1 legend “..” ← “.”

>> Revised.

Line 210 For instance, nad5 located ← add “is”  as : For instance, nad5 [is] located

>> Added.

Line 213-214 “As a result, the subtle differences in gene arrangement between these species will present new challenges to this conclusion.” - delete or rephrase better“ will present new challenges to this conclusion” this is a finding/result not a conclusion

>> Thanks for your comments. This sentence has been deleted.

line 287-288  “was clustered firstly clustered with the Diogenidae species Clibanarius infraspinatus and then clustered with another Diogenidae species Dardanus arrosor.” ← rephrase “clustered with the Diogenidae species Clibanarius infraspinatus and  Dardanus arrosor. However, the members of the family Coenobitidae … clustered within the same clade.”

>> This sentence has been revised as suggested.

line 292 “The ML bootstrap values supporting D. Edwardsii and Clibanarius infraspinatus as well as Dardanus arrosor were not high (<80), which may be explained by the different gene arrangements between D. Edwardsii and these two species, as discussed in part 3.1.” check his is not correct. The overall support for this clade is 100 but the branches or ordering of Clibanarius infraspinatus and Dardanus arrosor within the clade is not well supported, i.e. whether C. infraspinatus or D. arrosor is more related to D. Edwadsii than the other.

>> In the phylogenetic tree, the closely related species always cluster firstly. In the current study, C. infraspinatus or D. arrosor clustered with D. Edwadsii before any other species. Thus, C. infraspinatus or D. arrosor was more related to D. Edwadsii than the other species in spite that the branches of C. infraspinatus and within the clade were not well supported.

Line 294-295 “Thus, more species mitogenomes in the family Diogenidae still need to be investigated to fulfill the systematic phylogeny within the infraorder Anomura” – yes or nuclear markers. Even though better support would be achieved this is just the phylogeny of the mitogenomes which can differ from the nuclear genome and thus nuclear markes are also needed to reconstruct the phylogeny of the species.

>> Thanks for your constructive comments. As mentioned above, nuclear marks are useful for reconstructing the phylogeny of the species. However, as the title of this study “The complete mitochondrial genome of …”, the manuscript focuses on the systematic phylogeny within the infraorder Anomura based on mitochondrial genomes rather than the nuclear genomes.

Lines 298-302 “Comparing the nonsynonymous/synonymous substitution ratios (ω = dN/dS) has be299 come a useful approach for quantifying the impact of natural selection on molecular evo300 lution [26]. ω >1, = 1 and <1 indicate positive selection, neutrality and purifying selection, 301 respectively [52]. We examined potential positive selection in the Paguroidea lineage 302 (family Coenobitidae and Diogenidae in Fig. 3) using CodeML from the PAML package. “ true  this is in Methods and can be deleted here.

>> Deleted.

“no significant significance” – rephrase

>> It has been rephrased to “no significant difference”.

Conclusion- lines 350-351: “Our phylogenetic tree showed no controversies with the traditional morphological classification based on molecular methods that D. Edwardsii possessed the closest relationship with the species Clibanarius infraspinatus in the same family.” But Coenobitidae clustered within the Diogenidae family, suggesting that this is a single family.

>> Many thanks for the comments. This conclusion is somewhat in conflict with the results of the phylogenetic analysis. And the sentence has been changed into “Our phylogenetic tree based on molecular methods showed that D. edwardsii possessed the closest relationship with the species Clibanarius infraspinatus in the same family”.